# The Indirect Tribological Role of Carbon Nanotubes Stimulating Zinc Dithiophosphate Anti-Wear Film Formation

**DOI:** 10.3390/nano10071330

**Published:** 2020-07-08

**Authors:** Jarosław Kałużny, Andrzej Kulczycki, Wojciech Dzięgielewski, Adam Piasecki, Bartosz Gapiński, Michał Mendak, Tomasz Runka, Damian Łukawski, Oleksandr Stepanenko, Jerzy Merkisz, Krzysztof Kempa

**Affiliations:** 1Faculty of Civil and Transport Engineering, Poznan University of Technology, 60-965 Poznań, Poland; oleksandr.o.stepanenko@doctorate.put.poznan.pl (O.S.); jerzy.merkisz@put.poznan.pl (J.M.); 2Air Force Institute of Technology, 01-494 Warsaw, Poland; andrzej.kulczycki@itwl.pl (A.K.); wojciech.dziegielewski@itwl.pl (W.D.); 3Faculty of Materials Engineering and Technical Physics, Poznan University of Technology, 60-965 Poznań, Poland; adam.piasecki@put.poznan.pl (A.P.); tomasz.runka@put.poznan.pl (T.R.); damian.lukawski@put.poznan.pl (D.Ł.); 4Faculty of Mechanical Engineering, Poznan University of Technology, 60-965 Poznań, Poland; bartosz.gapinski@put.poznan.pl (B.G.); michal.mendak@put.poznan.pl (M.M.); 5Department of Physics, Boston College, Boston, MA 02467-3804, USA; kempa@bc.edu

**Keywords:** carbon nanotubes, anti-wear, tribology

## Abstract

Experimental studies reveal that the simultaneous addition of zinc dialkyl dithiophosphates (ZDDPs) and multi-wall carbon nanotubes (MWCNTs) to a poly-alpha-olefin base oil strongly reduces wear. In this paper, it is shown that MWCNTs promote the formation of an anti-wear (AW) layer on the metal surface that is much thicker than what ZDDPs can create as a sole additive. More importantly, the nanotubes’ action is indirect, i.e., MWCNTs neither mechanically nor structurally strengthen the AW film. A new mechanism for this effect is also proposed, which is supported by detailed tribometer results, friction track 3D-topography measurements, electron diffraction spectroscopy (EDS), and Raman spectroscopy. In this mechanism, MWCNTs mediate the transfer of both thermal and electric energy released on the metal surface in the friction process. As a result, this energy penetrates more deeply into the oil volume, thus extending the spatial range of tribochemical reactions involving ZDDPs.

## 1. Introduction

Carbon nanotubes (CNTs) have been studied intensely since they were first discovered in the 1990s [1]. However, only a relatively limited number of CNT applications has been developed so far [2,3,4]. Multi-wall carbon nanotubes (MWCNTs) can be grown in a relatively low-cost, catalytic chemical vapor deposition (CVD) process by using metallic nanoparticles as a catalyst [5]. The production costs of MWCNTs drop by a half every year [6,7], thus making many MWCNT applications increasingly more practical and cost-effective [4,8]. Single-wall carbon nanotubes (SWCNTs) are rolled, seamlessly connected graphene sheets and, as expected, exhibit anisotropic properties. Although their thermal conductivity is low perpendicular to their axis, it becomes very high along the axis (6000 W/(m·K)), which is several times greater than that of the diamond and more than ten times that of silver [2]. Their electric properties are also strongly anisotropic. Along the axis, SWCNTs can be either metallic or semiconducting, depending on their chirality (i.e., rolling direction), and in the metallic state they are characterized by excellent electrical conductivity, with a current density as high as 10^9^ A/cm^2^ [2]. This value is 1000 times greater than the maximum values obtained for conventional copper or silver. MWCNTs consist of multiple rolled graphene layers, thus they have larger diameters and are usually less pristine and uniform than SWCNTs. However, MWCNTs preserve some of the characteristics of SWCNTs (e.g., high anisotropy of the thermal and electrical parameters), although MWCNTs are exclusively metallic. Since the production of MWCNTs is much less expensive than that of SWCNTs, most commercially viable applications employ the use of MWCNTs.

MWCNTs have been widely tested and were found to modify the friction process [9,10,11]. Earlier experimental studies by the authors of this paper showed that the beneficial frictional behavior of MWCNTs can be observed only for well-dispersed CNTs that form agglomerate-free nanoliquids. This observation was first made indirectly during research on MWCNTs that covered a piston skirt. An impressive drop by as much as 16% in overall engine friction losses was measured after standard pistons were replaced by those covered with MWCNTs. This reduction was too large to be explained only by changes in friction released on the piston skirt. It was determined that the MWCNTs were abraded in small amounts from the piston surface into the oil during engine operation. These MWCNT fragments, perfectly dispersed in the engine oil due to the continuity of the abrasion process, were subsequently transported to all of the tested engine’s friction components [9,10,11]. A similar effect was also suggested in a paper by Reinert et al. [12]. In other experiments, it was observed that the addition of MWCNTs directly into engine lubricating oil caused a decrease in friction losses, but this was observable only after several minutes of engine operation had passed. At the same time, a significant decrease in the size of MWCNT-agglomerates was observed, which was caused by the shear stress occurring in the oil film during engine operation [13]. The presence of micron-sized and larger agglomerates may lead to a substantial increase in the friction and wear of an engine’s components [14]. In recent experimental studies, it was shown that large MWCNT agglomerates trapped in the converging lubrication gap blocked oil flow, thus clearly worsening friction conditions [15,16]. This effect might be attributed to all types of nanoadditives. In studies using CuO nanorods, it was observed that the nanorods’ dispersion stability and aspect ratio played an important role in their unique lubrication characteristics [17].

Depending on the mean pressure applied to the tribological system, two mechanisms of boundary layer formation by lubricating additives have extensively been described. The first mechanism is the adsorption or, more specifically, chemisorption of additive molecules on lubricated surfaces resulting in anti-wear (AW) film formation; the second mechanism is a chemical reaction of the additive with the material surface forming an extreme pressure (EP) layer. These mechanisms have been studied for many decades and have been extensively described in a large number of publications regarding tribochemistry [18,19], particularly for carbon-based tribofilms [20,21,22,23]. Under certain conditions, carbon nanostructures show superlubricity effects that result in the near vanishment in friction between the two sliding surfaces [24]. Carbon nanomaterials are prone to interact, in their tribological mechanisms, with oil molecules and with additives in particular, especially with ionic liquids [25,26,27,28].

The vast majority of experiments conducted to better understand tribological mechanisms use the lubricated surface characterization, which can only be performed after releasing mechanical and thermal loads in the tribometers. Various analytical methods allow us to characterize the durable reactions’ products with lubricating additives deposited on the metal surfaces of friction components. Both the chemical structure and the thickness of the deposited AW film are critical parameters necessary to protect lubricated surfaces from direct contact, i.e., against wear or seizing up.

Zinc dialkyl dithiophosphates (ZDDPs) are oil additives used in nearly all commercial lubricating oils that allow for the effective formation of AW films. The mechanism of ZDDP film formation was explained in a number of papers [29,30,31]. Generally, this mechanism can be significantly modified and improved in the presence of various chemical compounds added to the lubricating oil. More importantly, adding nanomaterials to oils may lead to new synergistic mechanisms of interaction with ZDDPs [23,32,33,34].

A mathematical model of heat-induced triboreactions and mechanical forces was proposed in papers by Kajdas et al. and Kulczycki et al. [35,36]. This model was developed to analyze the effectiveness of various lubricating additives and to study the effect of the base oil type on the activity of these additives. In this model, it is assumed that lubricated metal surfaces play the role of the solid tribocatalyst which emits electrons released in friction on single asperities. These electrons allow for an endothermic reaction of decomposition of the ZDDP molecules that are present in the oil—the products of this tribocatalytic reaction form durable deposits on the metal surface, i.e., the AW layer. In standard oil, electrons emitted on the metal surface have a limited penetrating distance inside the oil volume, thus limiting the efficiency of AW layer formation. The properties of MWCNTs, particularly their elongated shape and electrical conductivity, allow electrons emitted on the rubbing surfaces to deeply penetrate into the oil volume, thus enlarging the area of ZDDP reactions. In this indirect mechanism of tribological action, MWCNTs support strong AW film formation.

Until recently, it has been hypothesized that carbon nanomaterials dispersed in the lubricating oil provide the following: (i) purely mechanical action (rolling, local lifting of the lubricated surfaces and other mechanical interactions), (ii) a promotion of tribofilm formation on the lubricated metal surface (e.g., by CNT exfoliation), and (iii) the effect of metal surface polishing and mending [20]. Based on previous studies and experiments presented in this paper, we concluded that the role of MWCNTs in the friction process cannot be limited to these mechanisms only. **In this paper, we present experiments which clearly show that MWCNTs also modify friction indirectly by inducing a synergistic effect with ZDDPs.**

## 2. Materials and Methods

A High Frequency Reciprocating Rig (HFRR) tribometer fulfilling EN ISO 12 156-1 standards was used—Figure 1 shows the tribometer design and the tested surfaces (samples). In all of the experiments the load was set to 200 g, the stroke of the ball was set to 1 mm at a vibration frequency of 50 Hz and the oil temperature was maintained at 60 °C. The ball sliding on the plate surface was 6 mm in diameter, was produced using ISO 63-17-100 Cr6 steel, had a hardness of 58–66 HRC according to ISO 6508-1 standards, class G28. The plate material was also made of ISO 63-17-100 Cr6 steel annealed to obtain a hardness of 190 measured in the Vickers HV 30 procedure, according to norm ISO 6507-1. The yield stress of 100 Cr6 steel is 1700 MPa; the maximum contact stress according to the Hertz theory was 810 MPa. The plate surface was polished, with the R_a_ parameter less than 0.02 µm. Worn surfaces of plates tested in the HFRR tribometer were characterized using Raman spectroscopy, scanning electron microscope (SEM), and electron diffraction spectroscopy (EDS) techniques.

Industrial grade MWCNTs used in this research study were from Nanocyl (NC7000), prepared by catalytic chemical vapor deposition (CVD). Their average diameter and length were 9.5 nm and 1.5 µm, respectively. Carbon purity exceeded 90% and the BET surface area ranged from 250 to 300 m^2^/g. In this research, a CNT mass concentration in oil of 0.5% was used. Each 100 mL oil sample was prepared by adding CNTs to reach the specified concentration and sonicating it for about 30 min. Hielscher UP400St pin-type disintegrator was used in pulsed mode at the peak power of approx. 200 W. Only a portion of the CNTs could be well-dispersed in poly-alpha-olefin base oil (PAO), as it does not contain surfactants that are normally used in commercial lubricating oils. CNT agglomerates that did inevitably form in oil were filtered out after sonication by a single pass through a standard engine lubricating oil filter. For selected samples ZDDP was added in the last step to the dispersion of CNTs in oil. Pure base oil, without surfactants, was used to avoid the uncontrolled and undesirable impact that commercial oil additives have on observed ZDDP reactions. The same PAO6 oil was used for all of the HFRR tests; Table 1 shows this oil’s parameters. PAO6 is a commercially available synthetic oil with a hydrocarbon (isoparaffin) structure produced by catalytic oligomerization of linear α-olefins containing 8–12 carbon (including 1-decene) atoms in the chain. In order to maintain the experiment conditions as clear as possible, we did not use MWCNTs modified by attaching aliphatic chains, which had successfully supported dispersion in oils in our other tests. The actual concentration of MWCNTs dispersed and not agglomerated could not be exactly determined in the experiment—it is also not necessary for this study, in which the qualitative impact of MWCNTs on tribochemical reactions was analyzed. In cases where MWCNT dispersion is assisted by surfactants usually added to standard commercial lubricating oils, the necessary concentration of nanotubes can be radically decreased without reducing their function. Table 2 shows the composition of lubricants produced for the HFRR tests—these are pure PAO6 (sample A) for reference, followed by the same base oil enriched with CNTs (sample B) and ZDDP (sample C). Finally, we tested both CNTs and ZDDPs added to the oil simultaneously (samples D and E).

Raman spectroscopy measurements were conducted with a Via Renishaw Raman microscope using a 514.5 nm laser line at 5 mW energy. A lens with magnification ×50 (NA = 0.75) was used, and the CCD detector exposure time was set to 30 s. Spectral parameters were determined by Wire 3.0 software.

For SEM, a Tescan Vega 5135 microscope was used. EDS patterns of sulfur, zinc, phosphorus, and carbon were obtained by employing a PGT Avalon X-ray microanalyzer with a 55° take-off angle. The electron beam accelerating voltage was set at 8 kV to keep the sample penetration depth as small as possible.

Finally, tactile topography measurements of the friction track on the plates were conducted using a Hommel T8000 profilometer with a diamond tip (2 µm radius). For each plate, 1000 profiles perpendicular to the axis of the friction track were measured, each of them consisted of 4000 measuring points. The measured field was 2 mm × 4 mm, with vertical resolution at 2 nm. Due to the high peaks that were observed around the friction track on the reflecting metal surface, tactile topography turned out to be more reliable than the optical focus-variation microscope measurements.

## 3. Results and Discussion

The main experiment of this research study was the High Frequency Reciprocating Rig (HFRR) test [37]. The HFRR test is primarily dedicated to assessment of diesel fuel lubricity, but it is also useful for testing other lubricants [38]. During HFRR tests, both the dynamic friction coefficient and film thickness are simultaneously recorded. In this test, film thickness is measured indirectly via electrical resistance between the rubbing surfaces. Contrary to a sample examination performed first after completing the tribometer test, resistance measurement allows for in-situ observation of the dynamics of the AW film formation process. Numerous publications confirm a monotonic (but non-linear) relation between electrical resistance and the thickness of the tribo- or hydrodynamic film [39]. Figure 2 shows the friction coefficient (left panel) and electrical resistance between the rubbing surfaces (right panel) obtained during the HFRR test for four lubricants, namely, for base oil PAO6 (sample A), PAO6 with 0.5% MWCNTs (sample B), PAO6 with 1.5% ZDDPs (sample C), and PAO6 with 0.5% MWCNTs and 1.5% ZDDPs (samples D and E). The tribometer test time was set for 75 min for all samples except for sample E, in which the test was intentionally interrupted after 15 min, producing material for later spectroscopy that revealed anti-wear layer deposition dynamics. Each test was performed twice, showing excellent repeatability (see Table 3). Moreover, impressive repeatability of the HFRR test results can be observed when the curves of samples D and E are compared in Figure 1.

In Figure 2, in the left panel, samples A and B show a nearly identical, relatively large friction coefficient with an average of ca. 0.2. Significantly lower friction was observed for samples C and D, with an average value of only ca. 0.13. The right panel in Figure 2 shows electrical resistance measured between the rubbing surfaces, which monotonically correlates with the AW film thickness. As expected, the AW film had a negligible thickness for samples A and B. AW film thickness in sample C grew rapidly at first, up to 40% after 5 min, but then it decreased to nearly zero after 30 min of testing. A much more robust AW film was observed for the sample containing MWCNTs (sample D), except that it stabilized at a high thickness (100%) only after ca. 50 min. Based only on the parameters measured during the tribometer tests, it can be clearly concluded that the MWCNTs could not provide any substantial improvement when added to the PAO base oil as the sole additive. On the contrary, the MWCNTs revealed clearly visible tribological action in the same oil enriched with ZDDPs as the MWCNTs promoted the formation of a robust and thick AW film that was incomparably stronger to that built when the ZDDPs acted without the MWCNTs as an additive.

When analyzing the dynamics of wear and the AW layer deposition process, it should be noted that, particularly for the HFRR tribometer experiments, the load set at the beginning of the experiment remains constant, defining the constant force that presses the vibrating ball against the plate. During the tribometer test, the ball-to-plate contact surface increases due to wear, thus lowering local contact pressure—a parameter that must be regarded as critical for the AW film deposition. There is a minimum contact pressure that is necessary to produce enough energy allowing for ZDDP molecule decomposition which initiates the AW film formation. Conversely, a defined maximum contact pressure should not be exceeded, as this would suppress the deposition of ZDDP molecule decomposition products on the metal surface. We hypothesized that the presence of MWCNTs extended the window of AW film formation toward lower contact pressures by supporting the transfer of energy produced in the friction process on the metal surface to the oil volume, where ZDDP tribochemical reactions can be initiated. Due to wear and decreasing contact pressure, the AW film observed in sample C during the first few minutes of the HFRR test was removed after 30 min. In the same friction conditions, MWCNTs present in the oil allowed for extremely strong film formation (sample D, Figure 2, right panel), thus significantly decreasing the wear that was measured at the end of the experiment (Table 3).

In order to characterize the durable deposits in the friction track on the plates, each HFRR test was complemented with friction track topography (FTT), Raman spectroscopy, SEM imaging, and EDS mapping. Because of excellent HFRR test repeatability (Table 3), only samples from Trial 1 were considered for further characterization; Figure 3 presents an overview of the parameters describing the wear of the tested HFRR samples. For each HFRR sample plate, the groove (surface below the plate neutral level) and the elevation around it (surface above the plate neutral level) formed by the ball sliding on the plate surface were defined. Samples A and B are characterized by wide and deep grooves (Figure 3), as opposed to samples C, D, and E, which were lubricated by oil containing ZDDPs.

Figure 4 shows measurements using the tactile profilometer which confirm that the lubricant additives significantly affected groove topography. Steeper slopes are observed in the presence of ZDDPs as the only additive (sample C), while the addition of CNTs (samples D and E) resulted in the higher edge peaks that are clearly visible in Figure 5, which shows transverse central traces of the grooves for samples C, D, and E. The presence of CNTs in ZDDP-enriched oil caused a notable decrease of wear (sample D versus C, Figure 5), whereby the wear occurred primarily during the first minutes of the HFRR test, when AW film formation was unstable (compare samples D with E). This corresponds to the wear measured for the ball and plate, as presented in Figure 3. Due to the higher wear that was observed for sample C, the ball-to-plate contact surface increased, thus lowering the mean contact pressure. This was probably the reason for AW film degradation that started rapidly after the first 10 min of the HFRR test (Figure 2). In contrast, the lower wear observed for MWCNT-loaded sample D and the corresponding higher mean contact pressure resulted in the formation of a thick AW film.

Raman spectroscopy of the friction tracks was conducted to further clarify these conclusions. Figure 6 presents Raman spectra obtained for the MWCNT bulk material that was used as the oil additive in this study. The strongest Raman signal was observed at 1344, 1576, 2682, and 2917 cm^−1^, where the first two bands represent D and G modes of the CNTs.

Figure 7 shows Raman spectra for all of the samples, measured in the friction tracks. For both samples A and B, no C‒H stretching bands in the 2750–3000 cm^−1^ range were observed. CNT-specific D and G bands (occurring at 1340–1600 cm^−1^) were not found for plate B, which had been lubricated by CNT-enriched base oil. This explains the absence of either MWCNTs or carbon residues in the sample B friction track. A high intensity band in the 1300–1600 cm^−1^ range was observed for samples C, D, and E. This band is related to the presence of amorphous carbon. For these samples, hydrocarbon vibrations (2850–3000 cm^−1^) were also observed.

Table 4 lists the positions of the Raman bands, their origins, and corresponding references.

Finally, energy dispersive X-ray spectroscopy (EDS) was performed to map the samples (see Figure 8) and to identify the elemental deposits. As expected, phosphorus (P), sulfur (S), and zinc (Zn) deposits were found in the friction tracks on all of the plates lubricated with oil containing ZDDPs (samples C, D, and E). The elemental composition of the deposits strongly depended on the presence of CNTs in the oil. The addition of MWCNTs increased the concentrations of Zn and P in sample D, which is clearly visible when compared to sample C. In contrast, the friction track in sample C that was lubricated by oil without the nanotubes shows an enhanced S signal. The formation of AW film in CNT-containing oil reveals its dynamics when the EDS results are compared, obtained for sample E produced after 15 min of the tribometer test and for sample D which ran full time, i.e., the 75-min HFRR test. In the first friction stage, strong sulfur deposits were obtained (sample E) that were then suppressed by zinc (sample D). When compared to the dynamics of wear during the tribometer test, it can be concluded that the strong Zn deposits formed only in the presence of MWCNTs reduce the wear most efficiently (compare samples D and E in Figure 3, Figure 4 and Figure 5).

A weak carbon signal was observed in sample A and B’s friction tracks, but no increased carbon signal was observed in samples C, D, and E (see Figure 9).

We concluded that MWCNTs can reveal direct tribological action, which has been described in the literature extensively, though this action might be inefficient under certain friction conditions, such as the HFRR tests presented in this paper. On the contrary, under the conditions above, MWCNTs act indirectly by inducing a strong synergistic effect with the ZDDPs that are added to the base oil. An analysis of the dynamics of wear and deposit formation confirms our theory which was focused on the role of energy transfer in CNTs. We state that the observed ZDDP–CNT synergy is possible due to improved transfer of the electric energy that is generated in mechanical contacts of single asperities of the rubbing surfaces and distributed along the CNT axis in the oil film volume. Finally, the indirect anti-wear role of CNTs consists in extending the spatial range of the endothermic tribochemical reactions of standard oil additives such as ZDDPs, which stimulate strong AW film formation, away from the metal surface.

## 4. Conclusions

Our experiments show that the simultaneous addition of zinc dialkyl dithiophosphates (ZDDPs) and multi-wall carbon nanotubes (MWCNTs) to a poly-alpha-olefin (PAO) base oil significantly reduces wear. Since no CNTs were observed in the friction track deposits (both with EDS and Raman spectroscopy), we can assume that CNTs do not modify the lubricated surface directly but, instead, mediate the formation of a thick AW film that originates exclusively from the decomposition of ZDDP molecules. The CNTs present in the lubricating oil changed the Zn:P:S ratio in the ZDDP anti-wear layer remarkably by suppressing sulfur deposition and simultaneously promoting zinc and phosphorus deposits, thus significantly improving wear resistance.

In standard oil, the penetration distance of low-energy electrons emitted by a lubricated surface is small, thus they can initiate the reaction of the lubricating additives (e.g., ZDDPs) only in very close proximity to the metal surface. As a result, the protective AW layer is relatively thin and fragile. CNTs, due to their elongated shape and high thermal and electrical conductivity, mediate the transfer of energy released in mechanical contact of single asperities on the metal surface, thus conducting it away from that surface. Hence, this energy is distributed in a larger volume of the oil, which increases the region in which endothermic reactions can be activated and, ultimately, leads to the formation of a thick AW film.

This research study shows that adding MWCNTs allows us to reduce the concentration of ZDDPs while providing the same AW efficiency. This is an extremely valuable and practical result as, due to their environmental toxicity, the concentration of ZDDPs must be reduced in the newer generations of lubricating oils.

In summary, we conclude that the tribological roles of CNTs are not only limited to purely mechanical action on rubbing surfaces, as has generally been assumed. We believe that extending the knowledge on MWCNT tribological action is a prerequisite step that needs to be taken in order to bridge the gap between the compelling friction properties as observed in CNTs in laboratory conditions and their very limited practical applications. Furthermore, we presume that the acceleration of ZDDP anti-wear action as described in this paper is not the only indirect effect of CNTs in lubricants.

## Figures and Tables

**Figure 1 nanomaterials-10-01330-f001:**
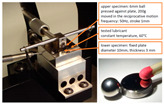
High Frequency Reciprocating Rig (HFRR) tribometer design, test conditions and friction track in the middle of the tested plate as pointed to by the tip of a match.

**Figure 2 nanomaterials-10-01330-f002:**
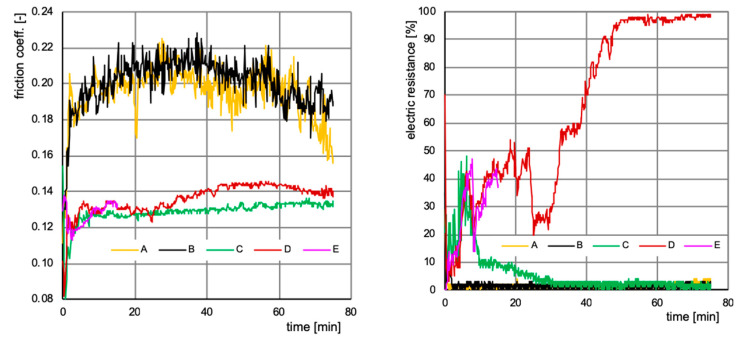
Dynamic friction coefficient (**left**) and resistance (**right**) between the rubbing surfaces measured during the HFRR test for samples A–E.

**Figure 3 nanomaterials-10-01330-f003:**
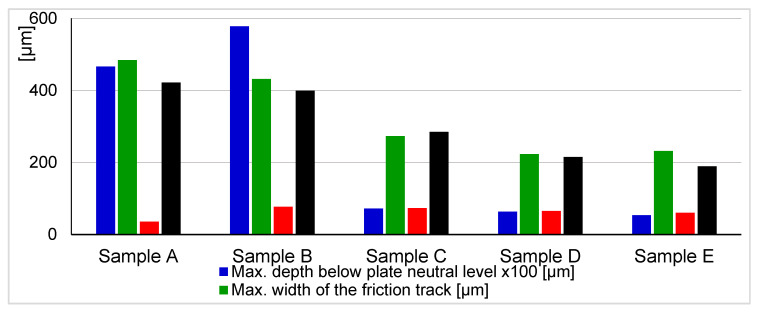
Key parameters describing friction wear of the samples.

**Figure 4 nanomaterials-10-01330-f004:**
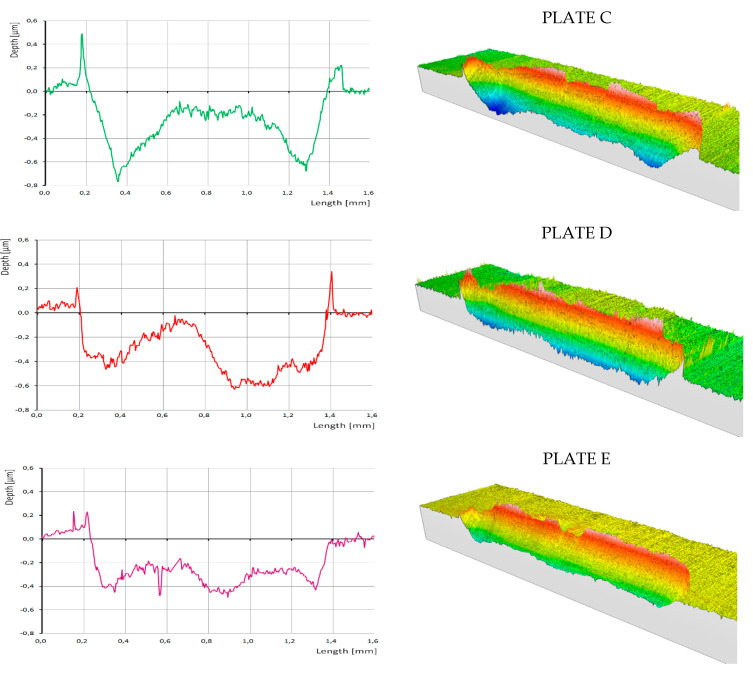
Longitudinal central trace of the friction track, as measured with a tactile profilometer (left), and the corresponding 3D topography image obtained from multiple longitudinal traces; a uniform scale bar and color assignment were used for all samples presented here.

**Figure 5 nanomaterials-10-01330-f005:**
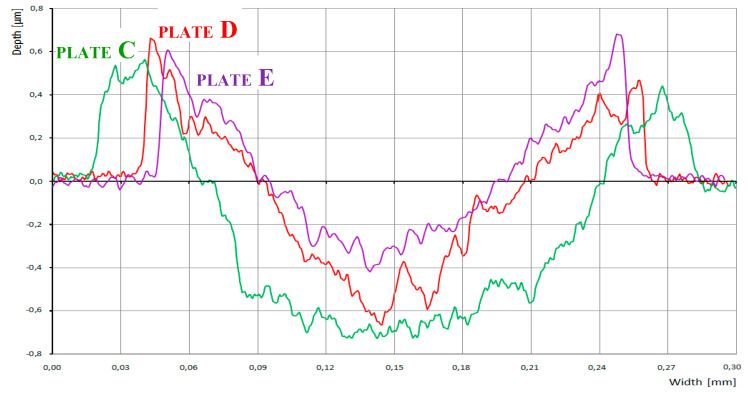
Transverse traces of the friction track measured by a tactile profilometer for samples C, D, and E.

**Figure 6 nanomaterials-10-01330-f006:**
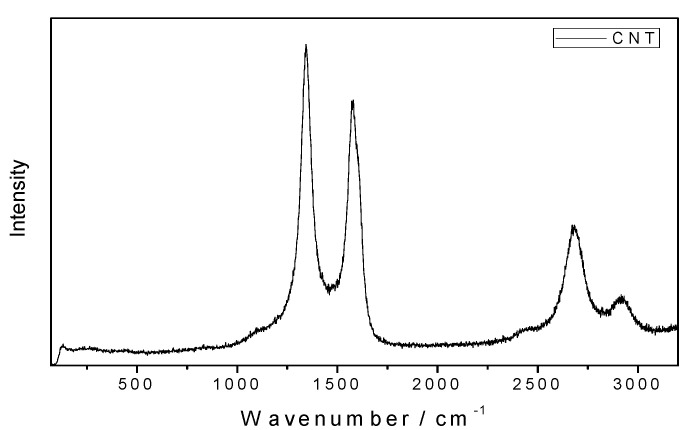
Raman spectra of the multi-wall carbon nanotubes (MWCNT) bulk material.

**Figure 7 nanomaterials-10-01330-f007:**
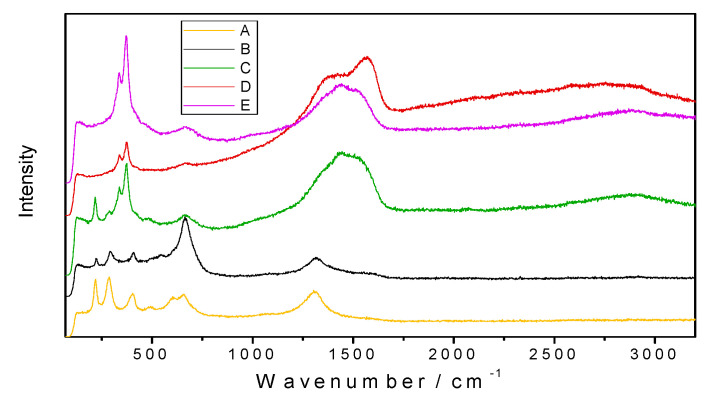
Raman spectra recorded from the friction tracks on plate samples A, B, C, D, and E.

**Figure 8 nanomaterials-10-01330-f008:**
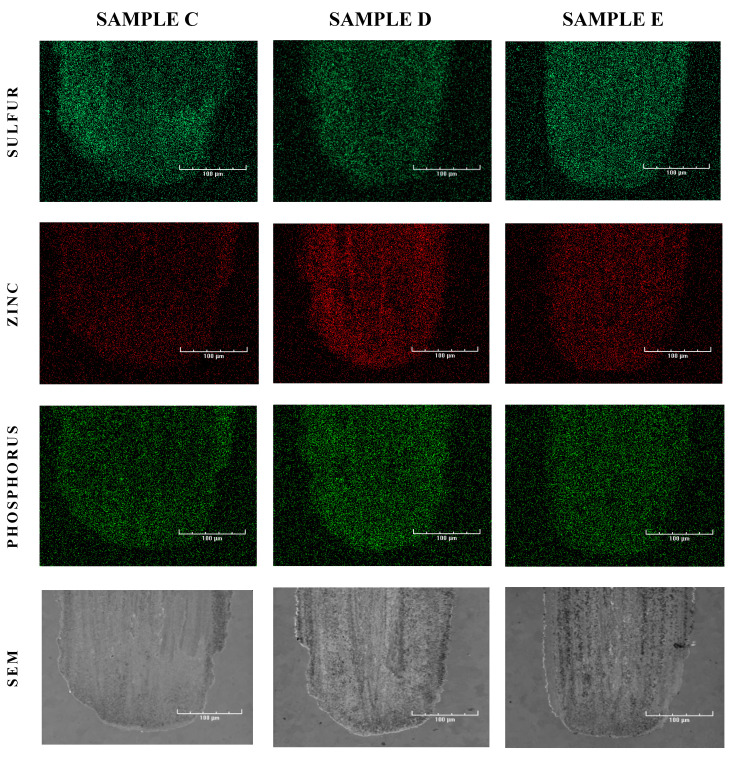
Electron diffraction spectroscopy (EDS) signal for sulfur, zinc and phosphorus in the friction tracks of sample plates C, D, and E, and the corresponding SEM images (bottom row).

**Figure 9 nanomaterials-10-01330-f009:**
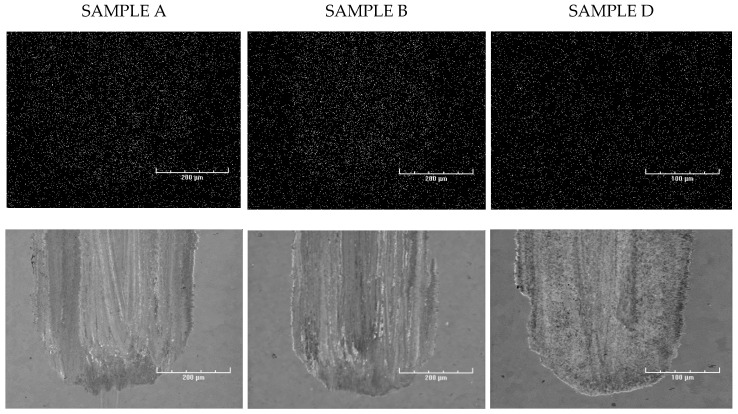
EDS carbon signal in the friction tracks of sample plates A, B, and D (top three images), and the corresponding SEM images below.

**Table 1 nanomaterials-10-01330-t001:** Parameters of PAO6 oil used as a reference and base oil to produce the tested lubricants.

Property	Unit	Value	Method
Specific gravity at 15.6 °C	g/cm^3^	0.827	ASTM D4052
Kinematic viscosity at 100 °C 40 °C −40 °C	mm^2^/s	5.8 31.0 7800	ASTM D445
Viscosity Index	-	138	ASTM D2270

**Table 2 nanomaterials-10-01330-t002:** Composition of PAO6-based lubricants used to produce samples in the HFRR tests.

Sample	CNT Concentration (wt %)	ZDDP Concentration (wt %)	HFRR Test Time (min)
A	0	0	75
B	0.5	0	75
C	0	1.5	75
D	0.5	1.5	75
E	0.5	1.5	15

**Table 3 nanomaterials-10-01330-t003:** Summary of the HFRR experiments.

Sample	Lubricant	Time (min)	Wear Scar Corrected (Ball) (µm)			Average Friction Coefficient (−)			Average Electrical Resistance (%) (Film Thickness)		
			Trial 1	Trial 2	Average	Trial 1	Trial 2	Average	Trial 1	Trial 2	Average
**A**	**PAO6**	75	422	432	**427**	0.195	0.178	**0.187**	1.5	3.2	**2.3**
**B**	PAO6 + CNT	75	399	404	**402**	0.201	0.197	**0.199**	1.5	1.4	**1.4**
**C**	PAO6 + ZDDP	75	285	273	**279**	0.129	0.132	**0.131**	5.9	7.4	**6.6**
**D**	PAO6 + CNT + ZDDP	75	215	208	**212**	0.136	0.135	**0.135**	64.5	76.0	**70.3**
**E**	PAO6 + CNT + ZDDP	15	189	193	**191**	0.127	0.132	**0.129**	28.0	35.1	**31.6**

**Table 4 nanomaterials-10-01330-t004:** Summary of the Raman experiments.

A	B	C	D	E	Assignment	References
Wavenumber (cm^−1^)						
286	294	286			FeO	[21,29]
		337	338	336	FeS_2_	[19,21]
		373	374	372	FeS_2_	[19,21]
494					α-Fe_2_O_3_	[21]
599					α-Fe_2_O_3_	[21]
659	668	669	662	663	Fe_3_O_4_	[19,20,21,29]
	1091				CaCO_3_	[19]
1307	1317				γ-FeOOH	[29]
		1362		1354	Carbon	[19]
	1554	1545			γ-Fe_2_O_3_	[19]
			1571	1575	Carbon	[19]
			2740	2720	C‒H	[29]
		2877			C‒H	[29]
			2930	2933	C‒H	[19,29]

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
