# Peer review of "The Indirect Tribological Role of Carbon Nanotubes Stimulating Zinc Dithiophosphate Anti-Wear Film Formation"

_nanomaterials, 2020, doi:10.3390/nano10071330_

Round 1

Reviewer 1 Report

Authors report an interesting study about the effect of addition of carbon nanotubes (MWCNT) on the wear behavior of poly-alpha-olefin (PAO) base oil modified with zinc dialkyl dithiophosphates, but major revisions are requested in order to satisfy the high impact factor of the journal.

Authors should confirm their conclusions not only by varying the loading of MWCNT between 0.5 and 1.5, but also to lower and higher loading in order to fix the loading range that make valid their conclusion. Furthermore, it could be very interesting perform a comparison among MWCNT with aspect ratio different from that used, that is 158 (1500/9.5).

Additional minor revisions are:

Materials and methods section: describe molecular formula and the main properties and supplier of PAO6

Materials and methods section: describe the details of the synthesis of the samples

Materials and methods section: insert a table with samples code and composition MWCNT and ZDDP wt%

Line 163-164: verify the sentence because sample B and D are the same (also for sample C and E)

Author Response

Reviewer 1

General remark:

Authors report an interesting study about the effect of addition of carbon nanotubes (MWCNT) on the wear behavior of poly-alpha-olefin (PAO) base oil modified with zinc dialkyl dithiophosphates, but major revisions are requested in order to satisfy the high impact factor of the journal.

Authors should confirm their conclusions not only by varying the loading of MWCNT between 0.5 and 1.5, but also to lower and higher loading in order to fix the loading range that make valid their conclusion. Furthermore, it could be very interesting perform a comparison among MWCNT with aspect ratio different from that used, that is 158 (1500/9.5).

The main idea of the manuscript is to confirm our theory regarding the indirect tribological mechanism of CNTs and its synergy with ZDDPs. Of course, in details these mechanisms will vary depending on the parameters. We are not able to include in this paper all of the possible variations of the friction conditions; even more, we will not break up the main concept of this manuscript into the particularities. We also did not test the CNTs’ mass concentration of 1.5% in the experiments presented in this study, as this would unavoidably turn the oil into a greasy substance [1]. Except for this extreme case, the CNTs’ concentration is not critical for the results in the friction processes since only a well-dispersed fraction of CNTs (ca. 0.01% or less) truly acts as a nanoadditive [2]. Despite any CNT-ZDDP interaction, the impact of CNTs’ morphology on friction is still not clear in detail. Indeed, we focused on this issue in our current research and we discovered that even if the CNTs’ diameter and length are the same, their tribological function may be completely different. Nevertheless, this is a topic that is complicated enough and demands a separate manuscript in the future. In the current manuscript we used the tribologically best CNTs according to our current knowledge.

The experiment conditions we chose allowed us to clearly confirm the indirect tribological mechanism of MWCNTs based on intensifying ZDDP action and, according to the best of our knowledge, this has never been described in the literature before.

In the revised version of the manuscript we added a short extension to the Conclusions section in order to express our idea even more clearly.

Additional minor revisions

Materials and methods section: describe molecular formula and the main properties and supplier of PAO6

The Materials and Methods chapter in the revised version was extended accordingly.

Materials and methods section: describe the details of the synthesis of the samples

The Materials and Methods chapter in the revised version was extended accordingly.

Materials and methods section: insert a table with samples code and composition MWCNT and ZDDP wt%

The Materials and Methods chapter in the revised version was extended accordingly.

Line 163-164: verify the sentence because sample B and D are the same (also for sample C and E)

The original text in lines 163-164 is: “(sample A), PAO6 with 0.5% MWCNTs (sample B), PAO6 with 1.5% ZDDPs (sample C), and PAO6 with 0.5% MWCNTs and 1.5% ZDDPs (samples D and E). The tribometer test time was set for 75 min”.

Sample D is sample B with additional 1.5% ZDDPs, thus it is not the same; similarly, sample E is sample C enriched with 0.5% MWCNTs. To avoid any confusion, Table 2 was added in the revised manuscript.

Reviewer 2 Report

This manuscript deals with the indirect tribological role of CNTs stimulating Zinc dithiophosphate anti-wear film formation. Followings are some comments regarding this manuscript.

  1. The authors used only one load value of 200g, 50 Hz, and 60 °C of oil temperature and all the results and conclusions are from the contact conditions. Why did the authors decide to use the only contact conditions?
  2. Oscillating travel distance is not given.
  3. Purpose and aim of contact conditions are not given in the manuscript. In other words, if the contact conditions vary, will AW film formation be still valid?
  4. Some important mechanical properties such as yield strength of the ball and plate should be provided. Also, what is the contact stress of 200 g (Hertzian)? Please compare the contact stress to the yield strength of 100Cr6.
  5. A photo or illustration of the test setup is needed for better understanding.
  6. In Fig. 2, wear of D and E are almost same in spite of run time difference. But, the worn trace of D shown in Fig. 3 looks pretty different from that of E. In the sample D, even no worn trace is shown between the lengths 0.6mm to 0.8mm. From a simple observation, the worn area of the D sample below zero plane is smaller than that of the E sample. Likewise, the selection of the wear track would result in different interpretation regarding wear.
  7. In fig. 4, the curve of E is purple and the legend is blue.
  8. The worn surface of the ball is not included. Also, wear of the plates is only discussed. For better wear behavior, wear of the balls are also included and discussed to support plate wear.
  9. In the line 259~260, the authors insist that “strong sulfur deposits were obtained (E) that were then suppressed by zinc (D)”. First, the meaning of the paragraph is not clear. From the sulfur EDS in the Fig. 7, sample D is richer in sulfur unlike the authors’ insistence.
  10. A more extensive chemical compounds analysis must be done using for example XPS rather than Raman analysis. As insisted, if the lubricant containing ZDDP is decomposed, there must be Zn, P, S element and there auxiliary compounds with O, Fe, C etc. It is well known that S with Fe reacts and forms a number of different iron sulfide compounds including FeS2. Among them, some are very good for wear resistance. So, it seems that this manuscript focused on only the role of carbon to explain anti-wear behavior. However, there may be other important anti wear compounds not related to carbon, and more examination should be done.
  11. This manuscript deals with anti-wear, but no wear experiments are done and discussed. If the anti-wear film is effective and promising, a number of wear tests must be done to validate authors’ findings.

Author Response

Reviewer 2

  1. The authors used only one load value of 200g, 50 Hz, and 60 °C of oil temperature and all the results and conclusions are from the contact conditions. Why did the authors decide to use the only contact conditions?

We chose the conditions with regard to norm requirements, keeping in mind that the results of our earlier experiments taught us how to ensure reproducible results. We obtained excellent reproducibility – which is clearly visible when comparing the values in Table 1 in the original manuscript: “Summary of the HFRR experiments”.

  1. Oscillating travel distance is not given.

We have corrected this – now the ball stroke is provided in the Materials and Methods section and was added to  Fig. 1.

  1. Purpose and aim of contact conditions are not given in the manuscript. In other words, if the contact conditions vary, will AW film formation be still valid?

As was mentioned in our reply to Question no. 1, our main purpose was to provide the experimental proveto our hypothesis of the tribological role of CNTs in energy transfer and ZDDP activation. ZDDPs acting as the sole additive can perform the anti-wear function in a certain range of friction conditions, particularly with regard to contact pressure, shear rate and temperature. As our experiment shows, the range of contact pressure allowing ZDDPs to conduct their typical anti-wear activity may be extended by adding MWCNTs to the oil (see line 222 and subsequent lines in the original manuscript). Definitely, MWCNTs and probably any other compound will be able to force ZDDP action in any theoretically possible conditions.

  1. Some important mechanical properties such as yield strength of the ball and plate should be provided. Also, what is the contact stress of 200 g (Hertzian)? Please compare the contact stress to the yield strength of 100Cr6.

These data are included in the extended Materials and Methods section.

  1. A photo or illustration of the test setup is needed for better understanding.

Please see Fig. 1 in the revised manuscript.

  1. In Fig. 2, wear of D and E are almost same in spite of run time difference. But, the worn trace of D shown in Fig. 3 looks pretty different from that of E. In the sample D, even no worn trace is shown between the lengths 0.6mm to 0.8mm. From a simple observation, the worn area of the D sample below zero plane is smaller than that of the E sample. Likewise, the selection of the wear track would result in different interpretation regarding wear.

We intentionally added various longitudinal and cross-sections presenting real 3D topography of the samples, which is rather complex. In the case of plates, a simple 2D cross-section can show some specific issues such as material deformation and deposit formation around the friction track, but this is insufficient for the final conclusions regarding wear. Much better global insight is provided by the 3D images in Fig. 4 (original manuscript) with uniform scale-color assignment for all of the presented samples. Sample D, which ran 75 minutes, shows generally higher wear than sample E that ran in the same conditions for only 15 minutes, however, this statement is not true for some parts of the selected cross-sections. Surely, this could happen since sample D is not sample E running an additional 60 minutes. We absolutely agree that the cross-sections should be chosen carefully. The longitudinal section always runs exactly in the middle of the friction track, and the cross-section was always chosen in the central part of the friction track, marked by high wear

In the revised version of the manuscript we added an explanation about uniform scale-color assignment.

  1. In fig. 4, the curve of E is purple and the legend is blue.

This has been corrected.

  1. The worn surface of the ball is not included. Also, wear of the plates is only discussed. For better wear behavior, wear of the balls are also included and discussed to support plate wear.

The wear of the ball and plate is quite well correlated – see Fig. 3 and Table 3 in the revised manuscript. We express this as follows: ”The presence of CNTs in ZDDP-enriched oil caused a notable decrease of wear (sample D versus C, Fig. 5), whereby the wear occurred primarily during the first minutes of the HFRR test, when AW film formation was unstable (compare samples D with E). This corresponds to the wear measured for the ball and plate, as presented in Fig. 3.” which are the same in the original and the revised manuscript.

  1. In the line 259~260, the authors insist that “strong sulfur deposits were obtained (E) that were then suppressed by zinc (D)”. First, the meaning of the paragraph is not clear. From the sulfur EDS in the Fig. 7, sample D is richer in sulfur unlike the authors’ insistence.

We can explain this comment only as an artifact resulting from an incompatibility in the picture presentation in the system used by the Reviewer. The thermal ZDDP decomposition products that are present around the friction track are nearly the same for sulfur as observed in samples C and D. Conversely, the friction track of sample E is clearly brighter than of sample D when comparing the maps for sulfur. We strongly advise to check the maps’ appearance carefully before printing/publishing.

  1. A more extensive chemical compounds analysis must be done using for example XPS rather than Raman analysis. As insisted, if the lubricant containing ZDDP is decomposed, there must be Zn, P, S element and there auxiliary compounds with O, Fe, C etc. It is well known that S with Fe reacts and forms a number of different iron sulfide compounds including FeS2. Among them, some are very good for wear resistance. So, it seems that this manuscript focused on only the role of carbon to explain anti-wear behavior. However, there may be other important anti wear compounds not related to carbon, and more examination should be done.

In the manuscript we compare the wear on samples lubricated with base oil enriched with ZDDPs and the same mixture to which MWCNTs were added. We can show clear wear reduction due to ZDDP-CNT synergy whereby CNTs added to the pure base oil without ZDDPs reveal no significant action. This is enough to document, for the first time, the indirect tribological role of MWCNTs and MWCNT-ZDDP synergy. EDS spectroscopy confirms the impact of MWCNTs on the elemental composition of the ZDDP film, whereby carbon is not directly included in this film. It always seems attractive to employ some additional methods such as XPS or XRD for friction track characterization. We have tried these, but our approaches failed due to the limited friction track surface length of ca. 1 mm and a width as small as 100 microns (see Fig. 1 in the revised manuscript). We did not exclude the possible extreme pressure (EP) actions and Fe-S chemical reactions which may occur simultaneously and widely independently of the MWCNTs’ effect on ZDDP AW film formation, which is the central point of this research and manuscript. We extended the Summary section in the revised manuscript accordingly.

  1. This manuscript deals with anti-wear, but no wear experiments are done and discussed. If the anti-wear film is effective and promising, a number of wear tests must be done to validate authors’ findings.

The focus and the real novelty of this manuscript is the revision and extension of existing models of MWCNT tribological action. It is quite disappointing that the number of industrial applications of MWCNTs is still very limited nearly 20 years after the first tribological experiments were conducted. Experiment results presented by authors working independently do not seems to fit one another. In our opinion, we first need to understand more deeply the tribological functions of MWCNTs, and that is the contribution our manuscript makes, by showing clearly new, or possibly only one of the new, mechanisms of CNT action.

[1] J. Kałużny, A. Merkisz-Guranowska, M. Giersig, K. Kempa. Lubricating performance of carbon nanotubes in internal combustion engines – engine test results for cnt enriched oil. Int J Auto Tech 2017; 18:1047-1059. https://doi.org/10.1007/s12239-017-0102-9

[2] Jarosław Kałużny; Marek Waligórski; Grzegorz M. Szymański; Jerzy Merkisz; Jacek Różański; Marek Nowicki; Mohanad Al Karawi; Krzysztof Kempa Reducing friction and engine vibrations with trace amounts of carbon nanotubes in the lubricating oil; Tribology International Pub Date: 2020-06-17 , DOI:  10.1016/j.triboint.2020.106484

Reviewer 3 Report

In the paper “The indirect tribological role of carbon nanotubes stimulating zinc dithiophosphate anti-wear film formation” (Manuscript ID nanomaterials-844979), the authors show that the simultaneous addition of zinc dialkyl dithiophosphates (ZDDPs) and multi-wall carbon nanotubes (MWCNTs) to a poly-alpha-olefin (PAO) base oil significantly reduces wear. The authors say with particular emphasis that MWCNTs act indirectly by inducing a strong synergistic effect with the ZDDPs that have been added  to the base oil. An analysis of the dynamics of wear and deposit formation confirms their theory which  was focused on the role of energy transfer in CNTs.

The advent of nanotechnology and nanotribology has expanded the research fields toward the use of nanoparticles or nanomaterials as additives in lubricating media like oil or grease. In this contest, recently, carbon nanotubes (CNTs) have emerged as a promising candidate for lubricant additive. Addition of CNT nanoparticles is well-known to improve anti-wear, extreme pressure, and load carrying capability of lubricating oil/grease. The superior tribological properties of CNTs have been attributed to their unique structure and dimension, which allow them to be easily active in the contact area and reduce interfacial friction. In my opinion, although the topic is interesting, the manuscript is not suitable for publication in Nanomaterials in its current form, because it shows many weaknesses. Even if some aspects are considered, they are not really new or studied in enough detail to provide new insights and significant information adequate to justify its publication.

Below, the main critical issues are listed.

  • The authors greatly stress the indirect action of carbon nanotubes as if to want to underline with this the true innovative aspect of their work. Indeed, the manuscript does not contain sufficient characterizations to effectively support the hypotheses they formulate and correlate the results obtained.
  • In the introduction, the continuous reference to SWCNTs is unnecessary since the research work is focused only on MWCNTs.
  • The keywords that need to be defined are missing.
  • The rheological characterization of the used lubricants which is essential to validate the results obtained has not been carried out. The rheology is an important aspect above all in light of what the authors asserted in the conclusions: “This research study shows that adding MWCNTs allows to reduce the concentration of ZDDPs while providing the same AW efficiency. This is an extremely valuable and practical result as, due to their environmental toxicity, the concentration of ZDDPs must be reduced in the newer generations 300 of lubricating oils.”
  • The samples listed in table 1 and the corresponding labels must be properly defined and described in the experimental part before being mentioned for the first time in the results and discussions section.
  • Many statements made by the authors do not provide a complete explanation of the observed phenomena.
  • English must be polished throughout the text.
  • The authors should emphasize how their study advances knowledge on the topic addressed in the manuscript.

Author Response

Reviewer 3

  1. The authors greatly stress the indirect action of carbon nanotubes as if to want to underline with this the true innovative aspect of their work. Indeed, the manuscript does not contain sufficient characterizations to effectively support the hypotheses they formulate and correlate the results obtained.

In the manuscript we compare the wear on samples lubricated with base oil enriched with ZDDPs and the same mixture to which MWCNTs have been added. We can show clear wear reduction due to the ZDDP-CNT synergy whereby CNTs added to the pure base oil without ZDDPs reveal no significant action. This is clear proof of the indirect tribological role of MWCNTs and MWCNT-ZDDP synergy.

We stress this effect since, according to the best of our knowledge, it was never described before and it can be a milestone towards industrial CNT application in lubricants.

  1. In the introduction, the continuous reference to SWCNTs is unnecessary since the research work is focused only on MWCNTs.

The phrase “SWCNT” appears six times in the original and only five times in the revised manuscript, exclusively in the first ten lines of the introduction. In the same ten lines the phrase “MWCNT” appears eight times. The reference to SWCNTs is necessary in the context of heat transfer and electric conductivity – which are the central point of our explanation for CNT-ZDDP synergy.

  1. The keywords that need to be defined are missing.

The keywords have been added to the revised version of the manuscript.

  1. The rheological characterization of the used lubricants which is essential to validate the results obtained has not been carried out. The rheology is an important aspect above all in light of what the authors asserted in the conclusions: “This research study shows that adding MWCNTs allows to reduce the concentration of ZDDPs while providing the same AW efficiency. This is an extremely valuable and practical result as, due to their environmental toxicity, the concentration of ZDDPs must be reduced in the newer generations 300 of lubricating oils.”

The rheological characterization could be added as well as many other feasible characterizations. This would not change our statement as presented in the answer to question 1 since it is generally known that adding 1.5% of ZDDPs to the base oil will not significantly change its rheology.

  1. The samples listed in table 1 and the corresponding labels must be properly defined and described in the experimental part before being mentioned for the first time in the results and discussions section.

The Materials and Methods section was extended to make the manuscript easier to read.

  1. Many statements made by the authors do not provide a complete explanation of the observed phenomena.

The general weakness of tribology is that the processes at a nano- and atomic scale are usually not accessible for in situ measurement. We usually measure macroscopic parameters such as friction and characterize wear and deposits first after finishing the friction process. Despite CNT-ZDDP synergy, some aspects of pure ZDDP film deposition mechanisms are still subject to discussion, even now, 80 years after the first ZDDP application was described.

  1. English must be polished throughout the text.

  1. The authors should emphasize how their study advances knowledge on the topic addressed in the manuscript.

We hope that this can be found in the revised manuscript.

Round 2

Reviewer 1 Report

Authors satisfied the reviewer's comments except for a more detailed description of the samples synthesis.

Author Response

Thank you for your acceptance for our manuscript. Regarding the sample preparation we added some details in the “Materials and Methods” section.

Reviewer 2 Report

It is all right that a number of the revisions have been made as suggested. Nonetheless, however, no clear reply or explanation regarding a few critical comments cannot be still found in the manuscript.

Thus, if possible, the authors are encouraged to give detailed revision result against each comment made during review. 

Author Response

First of all we are thankful for your careful study of our manuscript and numerous creative remarks which helped us to really improve the manuscript. Furthermore we are thankful for your acceptance for our vision of the manuscript which is focused on the indirect nature of the CNTs tribological action. We, authors certify this manuscript with our names and we believe the approach presented inside the manuscript may be fruitful invitation for re-thinking the nature of the CNTs roles in friction. Unfortunately at the moment we are not able to present research results with wider parameter variation and deeper than presented studies on the friction deposits characterization. We presume this is the focus of your actual critical remark. We still believe that our current manuscripts is complete as it is, at the same time we declare it clearly in the summary that it instead opens a new research direction. For example we absolutely don’t exclude CNTs impact on the EP additives typical action, we continue our research in the directions mentioned above.

Reviewer 3 Report

The authors sufficiently addressed the issues raised by the reviewer.

Author Response

Thank you for accepting our vision of the manuscript.